# Static Magnetic Field Inhibits Growth of *Escherichia coli* Colonies via Restriction of Carbon Source Utilization

**DOI:** 10.3390/cells11050827

**Published:** 2022-02-27

**Authors:** Haodong Li, Runnan Xie, Xiang Xu, Xingru Liao, Jiaxin Guo, Yanwen Fang, Zhicai Fang, Jirong Huang

**Affiliations:** 1Shanghai Key Laboratory of Plant Molecular Sciences, College of Life Sciences, Shanghai Normal University, Shanghai 200234, China; 1000479735@smail.shnu.edu.cn (H.L.); 1000497629@smail.shnu.edu.cn (R.X.); 1000459232@smail.shnu.edu.cn (X.X.); 1000497639@smail.shnu.edu.cn (X.L.); 1000479738@smail.shnu.edu.cn (J.G.); 2Heye Health Industrial Research Institute, Zhejiang Heye Health Technology, Anji, Huzhou 313300, China; yanwen.fang@heaye.com (Y.F.); zhicai.fang@heaye.com (Z.F.)

**Keywords:** magnetic field, carbon metabolism, free radicals, glycolate oxidase, long-chain fatty acids, oxidative stress, *Escherichia coli*

## Abstract

Magnetobiological effects on growth and virulence have been widely reported in *Escherichia coli* (*E. coli*). However, published results are quite varied and sometimes conflicting because the underlying mechanism remains unknown. Here, we reported that the application of 250 mT static magnetic field (SMF) significantly reduces the diameter of *E. coli* colony-forming units (CFUs) but has no impact on the number of CFUs. Transcriptomic analysis revealed that the inhibitory effect of SMF is attributed to differentially expressed genes (DEGs) primarily involved in carbon source utilization. Consistently, the addition of glycolate or glyoxylate to the culture media successfully restores the bacterial phenotype in SMF, and knockout mutants lacking glycolate oxidase are no longer sensitive to SMF. These results suggest that SMF treatment results in a decrease in glycolate oxidase activity. In addition, metabolomic assay showed that long-chain fatty acids (LCFA) accumulate while phosphatidylglycerol and middle-chain fatty acids decrease in the SMF-treated bacteria, suggesting that SMF inhibits LCFA degradation. Based on the published evidence together with ours derived from this study, we propose a model showing that free radicals generated by LCFA degradation are the primary target of SMF action, which triggers the bacterial oxidative stress response and ultimately leads to growth inhibition.

## 1. Introduction

The Earth has been generating a weak static magnetic field (SMF), also named the geomagnetic field (GMF), with an intensity around 50 μT at the surface for more than 4 billion years [1,2]. Since the origin of life occurred after the formation of GMF, it is likely that GMF information could have been explored by organisms as an important natural resource to tackle the problems encountered in evolution. Indeed, evidence is increasing to show that GMF can function as signals to help various organisms adapt to environmental changes and/or as energy to randomly affect biochemical reactions [3,4,5,6]. For example, many animals, including insects, turtles, fishes and birds, can congenitally migrate thousands of kilometers using GFM as a cue, and magnetotherapy is widely employed to alleviate pains and to treat a number of diseases, such as diabetes, immune disorders and mental disorders [6,7].

Single-cell microorganisms are extensively utilized to study series of magnetobiological effects and the underlying molecular mechanisms. To date, the best-understood mechanism of magnetosensation is the magnetotaxis of some bacteria, which can biosynthesize a fantastic magnetosensor that is composed of a nano-scale crystal of magnetite called a magnetosome [8]. However, it is hard to have common views on the nonspecific magnetobiological effects on microbe growth and the underlying mechanisms due to the diversity, complexity and inconsistency of the published data [9]. SMF effects are liable to be altered by many factors not only related to experimental designs, such as SMF intensity and gradients, bacterial strains, treatment time and culture conditions, but also to dynamic intracellular and extracellular targets of SMF [10,11]. Because bacteria exist ubiquitously in our surroundings and bodies, understanding the biological effects of SMF on bacteria will make a significant contribution to improving our environment, sanitation and health.

According to magnetic field strength, SMF is classified into the four subtypes, including the weak (<1 mT), moderate (1 mT–1 T), strong (1–5 T) and ultra-strong (>5 T) [12]. To date, available data on the interaction between moderate-intensity SMF and prokaryotic microbes were not consistent, whether positive, negative or null [13,14]. For instance, 30 min exposure to NdFeB magnets with an intensity from 45 mT to 3500 mT significantly reduced the viability of *E. coli* due to the damaged cell surface [15]. Similarly, 4 h treatment with 100 mT homogeneous SMF generated by an electromagnet significantly inhibited the adhesion of bacteria and consequent colony formation due to the disintegration of the cell membrane and the inner membrane release of the cytoplasm [16]. The Gram-negative *E. coli* was found to be more sensitive to SMF than the Gram-positive species *S. epidermidis* [16]. In some cases, the inhibition of bacterial growth by SMF was restricted within a certain time window during experimentation, called the biological window effect [17]. On the other hand, Potenza et al. (2004) reported that *E. coli* grew significantly faster in 300 mT SMF than in GMF when cultured in the modified liquid Luria–Bertani (LB) medium with 6 g/L glutamic acid and 4.5 g/L NaCl, whereas no magnetic effect was observed when the traditional LB medium was used [18]. In general, it is presumed that the higher the strength of the SMF, the worse the conditions for microbial growth and survival.

In contrast to there being a great deal of literature on magnetobiological phenotypes, our knowledge on the underlying mechanisms is very limited [6,19]. From a viewpoint of life evolution, moderate-intensity SMF is definitely a stress to all organisms living on the earth. It has been demonstrated that the activity of dehydrogenases and intracellular ATP levels, both of which are closely associated with bacterial stress responses, was significantly increased when bacteria were subject to SMF [20,21,22]. In addition, some stress marker genes, such as *relA* and *spot*, involved in the regulation of the level of guanosine penta- or tetra-phosphate (a pleiotropic regulator of stresses in bacteria), and *dnaK* and *groEL*, involved in helping macro-biomolecules’ folding and unfolding, were upregulated during the early stage of the SMF exposure [23,24]. Thus, one of the popular assumptions is that the interaction of moderate-intensity SMF with biological systems is mediated by reactive oxygen species (ROS), such as superoxide anion (O_2_^−^), singlet oxygen (^1^O_2_), hydrogen peroxide (H_2_O_2_) and hydroxyl radical (HO•), which can function as a signal to trigger signaling transduction and as a reagent to react with various biomolecules [25]. Since a large quantity of redox reactions take place constantly in organisms, ROS are inevitably produced in various subcellular structures. Based on the radical pair mechanism, SMF can lengthen the lifetime and increase the concentrations of free radicals, which possess unpaired electrons and a precessing magnetic moment, thus resulting in higher activity of radicals [26]. Indeed, SMF-treated cells were shown to produce more free radicals and express higher levels of antioxidant enzymes, including superoxide dismutase, catalase and peroxidase, which remove O_2_^−^, H_2_O_2_ and HO•, respectively [27,28]. Another interesting hypothesis is the enzymatic mechanism of magnetosensitivity, which states that intracellular enzymatic reactions accompanied with electron transfer are primary magnetoreceptors [19]. In the reaction center of some enzymes, the transfer of an electron from one molecular group to another generates an ion–radical pair (IRP) in the singlet state with antiparallel spin, which can be transformed into a triplet state with parallel spin by external magnetic fields and magnetic moments of surrounding nuclei via hyperfine interactions; consequently, the enzymatic activity is altered by the singlet–triplet conversion of the IRP in the active site [19]. The spin-dependent enzymatic reactions regulated by SMF or magnetic isotopes have been demonstrated to change the activity of enzymes involved in ATP and DNA biosynthesis both in vitro and in vivo [29,30,31].

Recently, more and more researchers are utilizing multiomics in combination with genetic, biochemical and molecular biology approaches to systematically investigate magnetobiological effects on organism growth and development. For example, three days of exposure of *Monascus ruber* M7 to 30 mT SMF significantly increased the accumulation of *Monascus* pigments and citrinin, which was consistent with the transcriptomic data showing that a number of genes involved in the pigment and citrinin biosynthesis, as well as in the primary metabolism, were upregulated by SMF [32]. In addition, moderate-intensity SMF could enhance the efficiency of a halotolerant yeast to decolorize azo dye via upregulating the expression of genes functioning in intracellular synthesis and the accumulation of glycerol [33]. These findings provide new and comprehensive insights into cellular mechanisms linking to the physiological phenotypes observed in SMF. Here, we showed that the exposure of *E. coli* cells to 250 mT SMF reduced the diameter of colonies. Transcriptomic and metabolomic analyses revealed that the inhibition of bacterial growth by SMF was attributed to the disturbance of carbon source utilization, particularly of long-chain fatty acid and glyoxylate metabolism, providing a metabolic basis for *E. coli* to adapt to the moderate-intensity SMF.

## 2. Materials and Methods

### 2.1. Bacterial Stains and Mutant Strain Construction

The *E. coli* TOP10 strain was used in the study. Bacterial knockout mutants Δ*glcD* and Δ*glcE* in the TOP10 background were constructed using the CRISPR-Cas9-based genome editing tool pCas/pTargetF system [34]. The sgRNA were designed based on the target genes. The editing templates had a 700 to 900 bp sequence homologous to each side (upstream or downstream) of the target genes. Once the target genes were deleted, two plasmids were cured to get the Δ*glcD* and Δ*glcE* knockout strains. The BW25113 line was obtained from Dr. Cao’s group at East China Normal University [35]. The Δ*glcD* and Δ*glcE* knockout strains in the BW25113 background were purchased from the KEIO collection and confirmed by PCR [36]. The data shown in this text were from the TOP10 genetic background unless otherwise stated.

### 2.2. SMF Treatments, Cell Culture and Measurement of Colonies

Permanent sintered magnets made from NdFeB containing PrNd (31 wt.%), B (1 wt.%), Fe (67.5 wt.%) and other (0.5 wt.%) (Hangzhou Permanent Magnet Group Co., Ltd., Xiaoshan, Hangzhou, China, www.china-hpmg.com, accessed on 25 February 2022) were used in this study. The magnet was a cube with each side having a length of 10 cm. A distribution map of the magnetic field was analyzed by FE-2100RD (Hunan Forever Elegance Technology, Hunan, China), and the data at each point were shown at XYZ axes (Figure 1B) [37]. The average magnetic intensity is 300 mT at the surface of the north and south poles. SMF intensity at the *E. coli* growth plane is 250 mT, as shown in Figure 1A. *E. coli* strains were first cultured in 5 mL liquid LB medium at 37 °C and 200 rpm overnight and then diluted with LB to obtain a series of gradient dilutions. Ten μL of bacteria solutions was inoculated on solid LB medium containing 1.5% bacto-agar. Glycolate or glyoxylate was added to the solid medium when it was necessary. The plates were put on the magnet pole as an SMF treatment or on an iron cube made of the same materials as that with the SMF but without a magnetic field as a control. The pictures were taken after bacteria grew on the solid medium for 10 h with a stereoscope (SDFPLAPO1xPF, Olympus, Japan). The CFUs were counted, and their diameters were measured with ImageJ. For transcriptome analysis, bacteria were cultured for 10 h and then collected from the solid plate, frozen in liquid nitrogen and stored at −80 °C. Samples for metabolome analysis were washed with PBS three times before being frozen in liquid nitrogen. For the measurement of bacteria growth curves, *E. coli* TOP10 strain (WT) and Δ*glcD* and Δ*glcE* mutant strains grew in 5 mL liquid LB medium until the value of OD600 reached 1, and then the bacteria were inoculated into fresh test tubes with 5 mL liquid LB medium and cultured at 37 °C with a 200 rpm shaking incubator. One hundred μL samples were then taken from the test tubes every one hour, and OD600 values were measured with a spectrophotometer.

### 2.3. Analysis of Transcriptome Sequencing and Quantitative PCR

Total RNA was extracted from samples using the TIANGEN RNA Extraction Kit. RNA quality and quantity were assessed by the NanoPhotometer^®^ spectrophotometer (IMPLEN, Westlake Village, CA, USA) and the RNA Nano 6000 Assay Kit of the Bioanalyzer 2100 system (Agilent Technologies, Santa Clara, CA, USA). RNA-seq library construction and sequencing were carried out according to the methods provided by Novogene Co., Ltd. (Beijing, China). Raw reads of the Fastq format were processed to obtain clean reads through removing those containing the adapter and poly-N and those with low quality. The clean reads with high quality were used for the subsequent analysis of gene expression.

The reference genome and gene model annotation files were downloaded from the website https://www.ncbi.nlm.nih.gov/assembly/ (accessed on 5 December 2020). The building of the index of and aligning of the clean reads to the reference genome were carried out with Bowtie2-2.2.3 [38]. HTSeq v0.6.1 (a software for high-throughput sequencing assay) was used to count the number of the reads mapped to each gene, and then FPKM of each gene was calculated based on the length of the gene and reads count mapped to this gene. Differential expression analysis between SMF and control groups (each with four biological replicates) was performed using the DESeq R package (1.18.0). DESeq provided differential expression using a model based on the negative binomial distribution. The resulting *p* values were adjusted using Benjamini and Hochberg’s approach for controlling the false discovery rate. Genes with an adjusted *p* value < 0.05 were assigned as differentially expressed. The statistical enrichment of differential expression genes in the Kyoto Encyclopedia of Genes and Genomes (KEGG) pathways was tested using the KEGG orthology-based annotation system (KOBAS) [39].

Validation of RNA-seq data was performed using quantitative real-time PCR (qPCR). Reverse transcription was carried out using 5 μg of RNA input for the iScript kit (Promega, A5001, Madison, WI, USA). qPCR was performed using SYBR Green Master Mix (Yeasen, China) on QuantStudio 3 (Thermo Fisher Scientific, Waltham, MA, USA) according to the instructions of the manufacturer. The level of 16S rRNA expression was used as an internal control, and the primers used are listed in Appendix A.

### 2.4. Metabolite Extraction and Profiling Analysis

To measure metabolite content, samples suspended with prechilled 80% methanol and 0.1% formic acid by vortex were sonificated for 6 min and then centrifuged at 5000 rpm at 4 °C for 1 min. The supernatants were freeze-dried and dissolved with 10% methanol. Metabolite profilings were analyzed by the LC-MS/MS system, in which a Vanquish UHPLC system (Thermo Fisher, Bremen, Germany) was coupled with an Orbitrap Q Exactive^TM^ HF mass spectrometer (Thermo Fisher, Novogene Co., Ltd., Beijing, China). The raw data were processed with regard to peak alignment, peak picking and quantitation for each metabolite using the Compound Discoverer 3.1 (CD3.1; Thermo Fisher). The normalized data were used to predict the molecular formula based on the additive ions, molecular ion peaks and fragment ions. Then, the peaks were matched with the mzCloud (https://www.mzcloud.org/, accessed on 6 April 2021), mzVault and MassList databases to obtain accurate qualitative and quantitative results.

The matched metabolites were annotated using the KEGG (https://www.genome.jp/ kegg/pathway.html, accessed on 11 April 2021), HMDB (https://hmdb.ca/metabolites, accessed on 11 April 2021) and LIPIDMaps databases (http://www.lipidmaps.org/, accessed on 11 April 2021). Principal component analysis (PCA) and partial least squares discriminant analysis (PLS-DA) were performed using metaX [40]. We employed univariate analysis (*t*-test) to calculate the statistical significance (*p* value). Metabolites with VIP > 1 and *p* value < 0.05 were considered differential metabolites. Volcano plots were used to filter the metabolites of interest based on the log_2_ FC and log_10_ (*p* value) of metabolites. These metabolites were mapped in metabolic pathways using the KEGG database.

## 3. Results

### 3.1. Moderate-Intensity SMF Inhibits Bacterial Growth

To investigate the effects of moderate-intensity SMF on bacterial growth, we treated *E. coli* with or without 250 mT SMF (Figure 1A). Ten μL of gradient dilutions was evenly spread on the surface of solid LB medium and then incubated at 37 °C for 10 h. SMF distribution of the horizontal plane at which the bacteria were grown was relatively uniform (Figure 1B). Our results showed that the size of the CFUs treated with SMF was apparently smaller than that of the control (Figure 1C), in spite of there being no difference in the number of CFUs between SMF-treated and control bacteria (Figure 1D). Thus, we measured the diameter of the CFUs, grouped the CFUs according to diameter and calculated the percentage of each group. The percentage of the diameter for SMF-treated CFUs reached a peak at 0.4 mm, which was 0.1 mm smaller than that (0.5 mm) of the control. The average diameter of SMF-treated colonies was significantly smaller than that of the control (Figure 1E). We also examined the dose-dependent effect of SMF on bacterial growth by adjusting the distance from the magnet to the bottom of the petri dish. Our results showed that the diameter of CFUs treated with 210 mT SMF was still significantly smaller than that of GMF control, while statistical difference disappeared when bacteria were exposed to 160 mT or 80 mT SMF (Figure 1F–H). Thus, our data suggest that there is a threshold of SMF intensity for the biological SMF effect on *E. coli* growth.

### 3.2. Transcriptomic Profiling Reveals That SMF-Regulated DEGs Are Enriched Mainly in Metabolism of Carbon Source Utilization and Stress Response

To dissect the biological processes regulated by the SMF, we performed high-throughput transcriptomic analysis of the bacteria treated with or without 250 mT SMF. Four independent biological samples were analyzed by RNA-seq, and the data are summarized in Figure 2A and Appendix A. A total of 4081 expressed genes were identified from all samples (Appendix A). Among the expressed genes, we found that 202 genes were differentially expressed between SMF-treated and control cells, with a *p* value lower than 0.05 (Appendix A). Further analysis showed that 71.8% (145) of the DEGs were downregulated while 28.2% (57) were upregulated by SMF. The maximal folds of the change in gene expression were 6.21 and 2.63 in the upregulated and downregulated genes, respectively. The PCA analysis showed that SMF treatment obviously altered the expression levels of the *E. coli* genes. The PC1 and PC2 principal components of the PCA plot accounted for 12.31% and 79.37% of the variation in the dataset, respectively (Figure 2B). These data suggest that 250 mT SMF is a weak or mild stimulus to *E. coli* with regard to gene expression, which is consistent with the weak phenotype observed in SMF-treated bacteria. The transcriptomic data were verified by quantitative real-time PCR (qPCR) analysis of the randomly selected 16 DEGs. In general, the qPCR results were in agreement with the transcriptomic data except for *yciH*, which was significantly upregulated in the RNA-seq but remained unchanged in the qPCR assay (Figure 2C).

To know the biological processes regulated by these DEGs in *E. coli*, we analyzed the functions of the DEGs with fold changes larger than 1.5 using KEGG. Our data showed that there were a total of 75 DEGs satisfied with our above demand, with 43 downregulated and 32 upregulated in SMF-treated bacteria (Appendix A). KEGG analysis revealed that these downregulated DEGs were significantly enriched in quorum sensing, the primary metabolism of amino acids, glyoxylate and dicarboxylate and fatty acids and the metabolism related to environmental stress (Figure 2D). The most downregulated genes were *glcD*, *glcE* and *glcF*, which encode the three subunits of glycolate oxidase, suggesting that SMF treatment significantly reduces the oxidation of glycolate into glyoxylate by glycolate oxidase. In addition, we noticed that SMF significantly downregulated the mRNA level of *glcB*, which encodes malate synthase G that catalyzes glyoxylate and acetyl-CoA into malate in the *glc* operon. These results indicate that carbon flux into gluconeogenesis and biomass production is reduced via the glycolate shunt that is partially overlapped with the tricarboxylic acid (TCA) cycle. SMF treatment also reduced mRNA levels of the *glp* operon genes involved in the catabolism of glycerol, such as *glpD*, *glpF*, *glpK* and *glpX*, which encode glycerol-3-phospate (G3P) dehydrogenase, glycerol diffusion facilitator, glycerol kinase and fructose-1,6-bisphosphatase, respectively [41]. Interestingly, several genes involved in fatty acid metabolism, including *fadI*, *fadJ* and *fadD*, which encode α subunit of the minor fatty acid oxidation multienzyme complex, β subunit of the minor fatty acid oxidation multienzyme complex and acyl-CoA synthetase, respectively, were downregulated by SMF treatment. These results suggest that several important carbon metabolic pathways, including glyoxylate, glycerate and fatty acids, are simultaneously downregulated by SMF. In amino acid metabolism, we found that all of the genes in the *astCADBE* operon, which encode the five enzymes of the ammonia-producing arginine succinyltransferase (AST) pathway required for the growth of *E. coli* with arginine as the nitrogen source, were downregulated in the SMF-treated bacteria, suggesting that exposure to moderate-intensity SMF alters nitrogen metabolism. Furthermore, eight (18% of the 43 downregulated DEGs) encoded proteins associated with the ATP-binding cassette (ABC) transporter system, including two dipeptide transporters (DppA/B), two putative D,D-dipeptide transporters (DdpA/B), autoinducer 2 (AI-2) transporter (LsrA) and a putative ABC transporter involved in natural transformation (YdcS/T/U). With regard to quorum sensing, *lsrB* encoding the AI-2 receptor, which is required for quorum sensing of *E. coli*, was concomitantly downregulated in the SMF-treatment cells. Thus, our data imply that SMF treatment probably affects bacterial growth via the quorum sensing mechanism, which has been demonstrated to play an important role in bacterial adaptation and colonization [42,43].

On the other hand, KEGG analysis indicated that the 32 upregulated DEGs with more than a 1.5-fold change were dominantly enriched in the aromatic amino acid biosynthesis pathway, pyrimidine metabolism and secondary metabolites biosynthesis. The most SMF-upregulated gene was *fliS*, encoding flagellar protein to regulate the length of flagellar filaments via facilitating flagellin export [44]. There were two other upregulated DEGs related to flagellar functions, namely *cheZ*, encoding a phosphatase to control the direction of flagellar rotation via dephosphorylation of signaling protein CheY in *E. coli* [45], and *flk*, with a role in flagellar phase variation related to bacterial virulence [46]. These results indicate that SMF affects flagellar function via flagellar development and signal transduction.

In SMF-treated bacteria, we found that two genes in the *trp* operon, *trpE* and *trpD*, which encode anthranilate synthase component I and II in tryptophan metabolism, respectively, and *yddG* encoding aromatic amino acid efflux transporter were significantly upregulated, indicating a possible increase in tryptophan biosynthesis. In addition, *thiE* and *thiC*, encoding thiamine phosphate synthase and phosphomethylpyrimidine synthase, respectively, were also upregulated by SMF treatment. TrpD was recently shown to be involved in thiamine biosynthesis via phosphoribosylamine formation in *E. coli* [47]. Interestingly, *purD*, encoding phosphoribosylamine-glycine ligase in purine biosynthesis, was upregulated in the SMF-treated cells. These data suggest that the biosynthesis of tryptophan, thiamine and purine is coordinately regulated. It was worth noting that 250 mT SMF treatment significantly enhanced the expression of a number of stress-related genes, such as two biofilm formation genes, *ymgA* (encoding a two-component system connector) [48] and *ykgJ* (encoding unknown protein) [49], *norW* (encoding an NADH: flavorubredoxin oxidoreductase to detoxify the free radical nitric oxide) [50] and *qorB* (encoding an NAD(P)H: quinone oxidoreductase). Taken together, our data suggest that the exposure of bacteria to SMF alters a series of biological processes, particularly those associated with stress, at the gene expression level.

### 3.3. Metabolomic Assays Show That Metabolites Significantly Altered in Content by SMF Are Enriched Mainly in Phospholipids, Fatty Acids and Aromatic Amino Acids

To assess the metabolic response of *E. coli* to SMF, we analyzed the metabolomes of 10 independent biological repeats treated with or without 250 mT SMF using the liquid chromatography–mass spectrometry (LC-MS) technology. Our untargeted metabolomic assay identified a total of 488 metabolites (Figure 3A; Appendix A and Appendix A), in which there were 55 differentially accumulated metabolites (DAMs) at the significant level (*p* ≤ 0.05 and VIP ≥ 1) between SMF-treated and control bacteria (Appendix A). The partial least squares discriminant analysis (PLS-DA) showed good separation (R^2^Y = 0.94, Q^2^Y = 0.35) of the SMF-treated samples from the controls (Figure 2B), suggesting that the bacterial metabolome is significantly altered by SMF.

Further analysis showed that these 55 DAMs (34 increased and 21 decreased) were enriched mainly in phospholipids, fatty acids, amino acids (AA) and derivatives from AA, in which the most dramatically altered compounds were phospholipids (Appendix A). Among the main phospholipids detected, the SMF treatment led to a significant decrease in phosphatidylglycerol (PG) but had no effect on phosphatidylethanolamine (PE), lysophosphatidylethanolamine (LPE), lysoPG (LPG) or lysophosphatidic acid (LPA) (Figure 3C; Appendix A). Detailed inspection revealed that PG (16:0/17:0) and phosphatidylethanol (PEtOH) (17:1/18:1) were the two most increased metabolites, with fold changes of 12.7 and 5.2, respectively, whereas the two most downregulated compounds were phosphatidylmethanol (PMeOH) (14:0/15:0) and PG (16:1/17:1), with fold changes of 0.33 and 0.46, respectively (Appendix A). In addition, other upregulated phospholipids were two PEs (PE (11:0/11:0) and PE (8:0/16:1)) and four LPEs (LPE 22:6, LPE 17:1, LPE 19:1 and LPE 20:4). These results indicate that moderate-intensity SMF leads to a mild compositional change in the membrane phospholipids of *E. coli*. Interestingly, the SMF-treated bacteria accumulated significantly higher levels of long-chain fatty acids (LCFA, with more than 18 carbons), except for stearic acid, but significantly lower levels of middle-chain fatty acids (MCFA, between 6 and 16 carbons) than the control (Figure 3C), suggesting that the degradation of LCFA is suppressed. This result is consistent with the RNA-seq data showing that the expression of *fadD*, encoding acyl-CoA synthetase, the first enzyme to degrade fatty acids in the inner membrane, is significantly downregulated by SMF. We found that in the enriched amino acids and their derivatives, *N*-acetyl tyrosine, serotonin, 2-oxindole, *S*-adenosyl methionine, valyl proline, pyroglutamic acid and nicotinuric acid were upregulated, while phenylalanine and γ-aminobutyric acid (GABA) were downregulated (Appendix A). These data are consistent with transcriptomic analysis showing that SMF treatment upregulated the expression of the genes involved in tryptophan metabolism, such as *trpD*, *trpE*, *mtr* and *yddG*, while it downregulated the expression of *aroG* and *aroH* genes for phenylalanine metabolism. In addition, other metabolites that accumulated relatively more in the SMF-treated cells than the control were related to nucleotide and nitrile metabolism (Appendix A). Taken together, our metabolomic data demonstrate that SMF can modify the bacterial metabolism of phospholipids, fatty acids and amino acids.

### 3.4. Both Glycolate and Glyoxylate Rescue the SMF-Inhibited Growth Phenotype

Based on our RNA-seq data showing that *glcD* and *glcE* were the most downregulated DEGs by SMF, we proposed that SMF probably inhibits cell growth via the glyoxylate bypass. To test this hypothesis, we cultured bacteria on the solid medium supplemented with various concentrations of glyoxylate. Our results showed that the difference in bacterial growth between SMF-treated and -untreated bacteria was gradually narrowed down with the increase in glyoxylate concentrations, and it completely disappeared when the glyoxylate concentration reached 5 μM (Figure 4A). We also tested whether glycolate, the substrate of glycolate oxidase, could mimic the effect of glyoxylate on bacterial growth in SMF. As expected, the addition of glycolate to the media also recovered the growth phenotype of SMF-treated bacteria (Figure 4B). However, the glycolate concentration that completely suppresses the growth inhibition of SMF was higher than that of glyoxylate (Figure 4A,B). Thus, our results suggest that both glycolate and glyoxylate can recover the bacterial growth in 250 mT SMF.

To genetically investigate the role of glycolate oxidase in regulating bacterial growth under SMF, we constructed the knockout mutants Δ*glcD* and Δ*glcE*. Our results showed that there was no significant difference in the diameter of the mutant clones between SMF-treated and -untreated cells (Figure 4C,D), suggesting that knockout mutant strains of *glcD* and *glcE* are insensitive to SMF. In addition, the two mutants, Δ*glcD* and Δ*glcE*, grew significantly slower than the wild type under GMF conditions (Figure 4E), suggesting that glycolate oxidase is not specific for *E. coli* adaptation to the SMF stimulus. Taken together, our data demonstrate that glycolate oxidase is a limiting factor to mediate the inhibitory effect of SMF on *E. coli* growth.

## 4. Discussion

In this study, we found that 250 mT or 210 mT SMF treatment can significantly reduce the size but not the number of *E. coli* colonies grown on solid LB medium. This phenotype is generally consistent with the popular opinion that moderate-intensity SMF negatively affects bacterial growth and survival [16,25,51,52]. In contrast to the liquid culture system, the utilization of a solid culture medium can keep all bacterial cells in a relatively stable and equal magnetic field. Under our conditions, the inhibitory effect of SMF on colony growth disappeared when the intensity was reduced to 160 mT, which is 2 cm away from the surface of the 300 mT magnet, suggesting that bacteria can resist a certain level of the moderate SMF stress and that there is a threshold of SMF intensity for biological effects. Thus, the intensity of SMF, particularly generated from a permanent magnet whose strength decays very quickly in the vertical direction of the two poles, should be taken into account in biological experiment designing. In addition, our RNA-seq and untargeted metabolomic assays revealed that 250 mT SMF only altered about 5% and 11% of the totally expressed genes and metabolites, respectively. In yeast, analysis of the SMF effect on global gene expression also showed that about 5% of the total genes were differentially expressed in the group treated with 206.3 mT SMF, compared to the control [33]. If bacteria are grown in weaker SMF intensity, such as 160 mT and 80 mT, both the number of DEGs and the level of DEGs expression will be dramatically reduced, which may have led to no biological effects on bacterial growth in our study. We therefore conclude that moderate-intensity SMF is a weak or mild stimulus, and fine control of all experimental conditions is a critical prerequisite to obtain stable and consistent results on magnetobiological effects.

Transcriptomic analysis revealed that the slow growth of SMF-treated bacteria was closely related to downregulated expression of the *glc* operon involved in central carbon metabolism. Glycolate oxidization catalyzed by glycolate oxidase is the first step of glycolate utilization, and glyoxylate is an important intermediate metabolite in the glyoxylate bypass/shunt of the central metabolism [53]. Our subsequent assays with physiological and genetic approaches confirmed the role of glycolate oxidase in SMF-induced growth inhibition. First, the addition of either the substrate or the product of glycolate oxidase to the solid LB medium successfully removed the inhibitory effect of SMF on bacterial colony growth, suggesting that glycolate or glyoxylate is a limiting carbon source for bacterial growth in SMF. Second, the null Δ*glcD* and Δ*glcE* mutants were not sensitive to 250 mT SMF anymore. Since Δ*glcD* and Δ*glcE* bacteria also grew more slowly than the wild type under the normal GMF condition, glycolate oxidase is not specific for the SMF stimulus. It is an interesting question why the glycolate pathway is dramatically suppressed by SMF. Glyoxylate formed by glycolate oxidase can be further metabolized by glyoxylate carboligase or malate synthase (MS), which direct glyoxylate into the glycolic pathway or the TCA cycle, respectively. It seems that the glyoxylate bypass is suppressed by SMF, because expression of *glcB* was coordinately downregulated with *glcDEF* genes. It is well characterized that the glyoxylate bypass plays an important role in balancing carbon resources between the oxidative steps of the TCA cycle and the provision of carbon skeleton molecules for gluconeogenesis when *E. coli* is grown on a medium containing acetate, fatty acids or ketogenic amino acids as a sole carbon source [54]. Based on our RNA-seq data that SMF treatment did not alter the expression of the genes involved in the TCA cycle or the first dedicated enzyme isocitrate lyase in the glyoxylate bypass, we propose that the physiological significance of the decreased expression of glycolate oxidase lies in reducing carbon flux to biomass production and in the maintenance of energy supply via the TCA cycle. This hypothesis is in agreement with the current knowledge on our understandings of flux control at the branch point of the TCA cycle and the glyoxylate bypass. The common substrate of the two pathway is isocitrate, which is catalyzed by NADP-dependent isocitrate dehydrogenase (ICD) in the TCA cycle and by isocitrate lyase in the glyoxylate shunt. ICD activity is primarily controlled by the bifunctional enzyme AceK, which specifically phosphorylates and dephosphorylates ICD to inactivate and activate its activity, respectively [54]. Interestingly, the kinase activity of AceK is allosterically inhibited by glyoxylate [54]. Under our experimental conditions, the reduced expression of glycolate oxidase led to a lower level of its product, glyoxylate, facilitating AceK to dephosphorylate ICD and thus to increase ICD activity. With regard to the glyoxylate bypass pathway, since expression of the *glc* operon was reported to be induced by glycolate via the constitutively expressed factor GlcC [53,55], the lower level of glycolate results in a significant decrease in the expression of the *glc* operon in SMF-treated cells. Thus, our data demonstrate that glycolate oxidase plays an important role in bacterial adaptation to moderate-intensity SMF.

Besides the *glc* operon, we found that two other operons, *glpFKX* (responsible for glycerol catabolism) and *astCADBE* (for arginine catabolism), were also significantly downregulated in the SMF-treated bacteria. Since the AST pathway provides nitrogen as well as succinate, an intermediate of the TCA cycle, the expression of the *ast* operon is dynamically regulated by nitrogen and carbon starvation via coordinated actions of multiple transcription factors [56]. Interestingly, the *ast* operon promoter was strongly induced by glycerol in exponentially growing cells when arginine was used as a nitrogen source [56], indicative of a highly coordinated expression of the *glp* and *ast* operons. The mechanism by which the glyoxylate, glycerate and AST pathways are coordinately regulated by SMF will be an important topic to investigate in the future.

The membranes of *E. coli* are composed of three major phospholipids, PE (about 75% of membrane lipids), PG (about 20%) and cardiolipin (CL, about 3%); minor phospholipids, such as LPE, phosphatidylinositol, phosphatidic acid and phosphatidylserine; and phosphorus-free membrane lipids [57]. The phospholipid composition and content of the inner and outer membranes can be modified by environmental changes via replacing the fatty acid moieties as well as the head group. In general, our metabolomic analysis showed that SMF treatment caused phospholipid modification to occur in both the fatty acids and the head group. In 250 mT SMF-treated cells, the amount of PG and PE with saturated fatty acids increased, while the PG with unsaturated fatty acids decreased, indicative of a decrease in membrane fluidity; in addition, SMF treatment increased lyso-PE, which might be produced from existing PE by phospholipase and has the advantage of quickly responding to environmental changes. Considering that RNA-seq analysis did not demonstrate that genes involved in membrane lipid modifications were significantly up- or downregulated by SMF, it is likely that membrane phospholipid modulation is under control at posttranscriptional levels.

LCFA can act as an energy-rich nutrient source, an essential substrate for membrane lipid biosynthesis and a precursor for signal molecules, regulating a series of cellular processes. In *E. coli*, exogenous LCFA are transported across the outer membrane by a β-barrel outer membrane protein, FadL, and subsequently catalyzed by in an inner-membrane-associated acyl-CoA synthetase, FadD. FadD is also required for the activation of endogenous LCFA released from membrane lipids [58]. Mutations in *fadD* lead to the failure of *E. coli* to grow on the media containing LCFA as a sole carbon source and to the accumulation of fatty acids inside the cell [58]. We found that in SMF-treated bacteria, *fadD*, together with *fadI* and *fadJ*, was significantly downregulated, which is in agreement with our metabolomic data that LCFA accumulated while the intermediates (MCFA) in the β-oxidation cycle decreased. Taken together, our data suggest that membrane phospholipid remodeling is associated with LCFA catabolism during the adaptation of *E. coli* cells to SMF.

Both RNA-seq and metabolomic assays showed significant changes in the metabolism of aromatic amino acids in 250 mT SMF-treated *E. coli*. In SMF-treated bacteria, *trpD* and *trpE* were upregulated. It is well documented that the expression of the *trp* operon is suppressed by the tryptophan-activated *trp* repressor and activated by uncharged tRNA^Trp^ accumulation [59,60]. In addition, *yddG*, encoding aromatic amino acid efflux transporter, was upregulated under the SMF condition. These results suggest that intracellular tryptophan content is lower under the SMF conditions than the control due to more tryptophan being exported outside the cell or assimilated. In *E. coli*, tryptophan can be degraded into indole, pyruvate and ammonia or can be further turned into many other biomolecules, such as serotonin. Our metabolomic data indeed revealed that the levels of tryptophan-derived metabolites, such as 2-oxindole and serotonin, were increased in the SMF-treated bacteria. Interestingly, tryptophan and indole are involved in quorum sensing and preventing biofilm formation, suggesting that SMF treatment affects biofilm formation [61,62].

Bacteria secrete many molecules facilitating cell-to-cell communication (quorum sensing, QS), by which they can react accurately to the ever-changing environment [42]. QS, which involves the production, release and community-wide detection of small hormone-like molecules termed autoinducers (AI), plays an important role in regulating gene expression and biofilm formation [42,63,64,65,66]. One of the best-characterized QSs is the LuxS/AI-2 system, in which LuxS is the key enzyme to producing AI-2 from the precursor *S*-adenosylmethionine, while AI-2 accumulates extracellularly, peaks in the exponential phase and then declines rapidly in the stationary phase. The reduction in the extracellular AI-2 is attributed to transportation into the cells via the transporter system encoded by the *lsrACDBFG* operon [67,68]. *lsr* expression is regulated by the cyclic AMP (cAMP)/cyclic AMP receptor protein (CRP) complex (cAMP–CRP) and two factors LsrK and LsrR encoded by the *lsrRK* operon [69]. Based on the transcriptome data, we found that *lsrA* and *lsrB* involved in AI-2 uptake were significantly downregulated by SMF. The lowered levels of *lsrA* and *lsrB* transcripts reflects a higher activity of the luxS/AI-2 system, because strong AI-2 signaling inhibits *lsrB* expression via LsrR [70,71]. Consistently, accumulated *S*-adenosylmethionine was higher, probably leading to more AI-2 under SMF, compared to the control. In addition, *lsr* expression is suppressed by preventing the activation of the cAMP/CRP complex due to the G3P accumulation in *clpD* mutants [72]. Interestingly, our RNA-seq data showed that *glpD* was significantly downregulated by SMF. It is plausible to assume that the cAMP–CRP-dependent mechanism plays an important role in coordinating the expression of *lsr* and *glp* operons in the SMF-treated bacteria. Thus, our data suggest that SMF treatment delays bacterial entry into the stationary phase.

To date, the radical pair mechanism has been well recognized and widely testified as a good model to explain a variety of SMF bioeffects [25,26,27]. Free radicals, such as ROS and reactive nitrogen species, are inevitably produced from redox reactions in organisms. SMF can increase the lifetime and concentration of radical pairs [26]. Such effects are more pronounced when the radicals are generated in some physical boundary, such as micelles and bilayer membranes, due to their crowding together [25]. Recent studies showed that LCFA utilization generates higher levels of ROS due to the lack of the antioxidant ubiquinone, a lipid-soluble electron carrier in the electron transport chain (ETC), compared to glucose, succinate or acetate carbon source [73]. This is because a large number of reduced cofactors, including NADH and FADH2, are generated in LCFA degradation, thus conferring redox stress and electron leakage that occurred in the inner membrane of *E. coli*. In this study, we found that LCFA utilization was inhibited to a certain degree, and the expression of some genes involved in defense against redox stress were coordinately altered under moderate-intensity SMF. Thus, when *E. coli* is subjected to SMF, LCFA-induced ROS accumulate in the inner membrane of *E. coli* and trigger a bacterial response to ROS stress.

Here, we propose a simple model to explain the growth inhibition of *E. coli* colonies by SMF based on the available evidence together with that derived from this study (Figure 5). Under the normal GMF condition, *E. coli* cells keep homeostasis between growth and stress resistance by utilizing various carbon sources. In contrast, under the middle-intensity SMF condition, a number of carbon metabolic pathways, particularly for LCFA and glyoxylate utilization, are significantly reduced, while some stress-related metabolic pathways, including the synthesis of aromatic amino acids, AI-2 and antioxidants, increase. Free radicals generated by LCFA utilization in the inner membrane of the *E. coli* envelope could be the primary target of SMF action, leading to an increase in ROS levels and the generation of redox stress. Then, ROS stress stimulates a series of sophisticated defense responses to maintain redox homeostasis via reprogramming related gene expression. For example, LCFA degradation is reduced due to downregulated expression of *FadD*, probably via a feedback inhibition mechanism; less of the carbon source is channeled into gluconeogenesis via reducing the glyoxylate bypass (and probably the glycerate pathway), while more nutrients are allotted to the pathways for the biosynthesis of antioxidants and biofilm. Integrative analyses of the RNA-seq and metabolomic data imply that the cAMP/CRP complex and the LuxS/AI-2 QS system play an important role in concerted regulation of gene expression. In addition, we cannot exclude the possibility that the observed biological effect is caused by direct influences of SMF on enzymatic activity via the formation of a magnetic-field-sensitive ion–radical pair in the reaction center. In general, we propose that the application of moderate-intensity SMF to *E. coli* mimics a mild redox stress that inhibits colony growth. In the future, it will be interesting to test the hypothesis that free radicals generated by LCFA degradation are primary magnetosensors that mediate biological SMF effects in *E. coli*.

## Figures and Tables

**Figure 1 cells-11-00827-f001:**
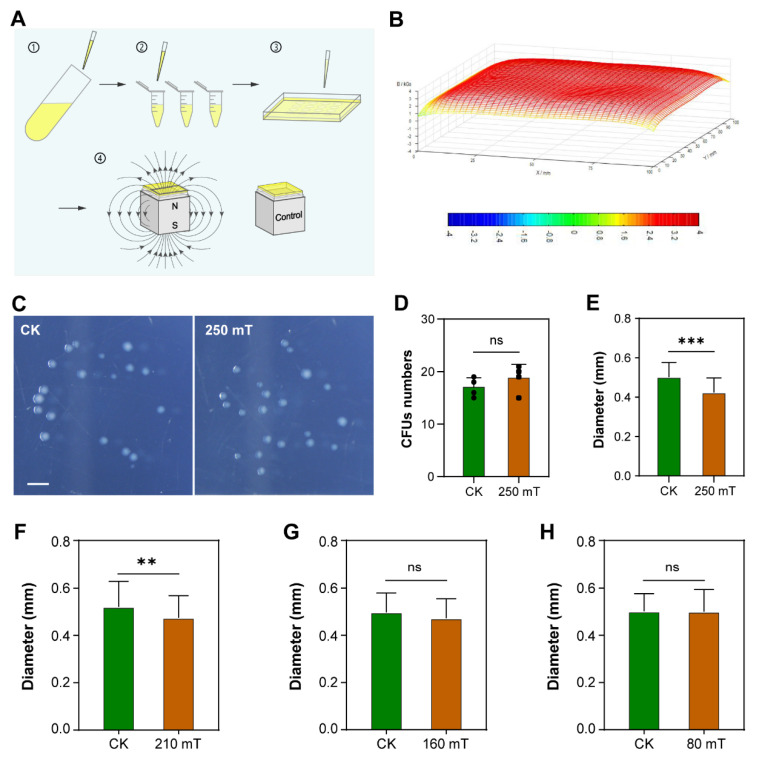
SMF treatment inhibits bacterial growth. (**A**) A diagram to show experimental procedure. Five milliliters bacteria cultured overnight were diluted step by step, and the dilution was spread on the surface of the medium. The petri dish was put on a magnet or a cubic iron as a control at 37 °C. (**B**) Magnetic field distribution of the *E. coli* growth plane, 10 mm above the NdFeB magnet. (**C**) Phenotype of colonies treated with (250 mT) or without (CK) SMF. (**D**) Effect of SMF on the number of CFUs. (**E**–**H**) Effect of SMF on the diameter of CFUs. Data shown as mean ± SD. *n* > 50. Statistical analysis was performed by *t*-test. ^ns^
*p* > 0.05, ** *p* < 0.01, *** *p* < 0.001.

**Figure 2 cells-11-00827-f002:**
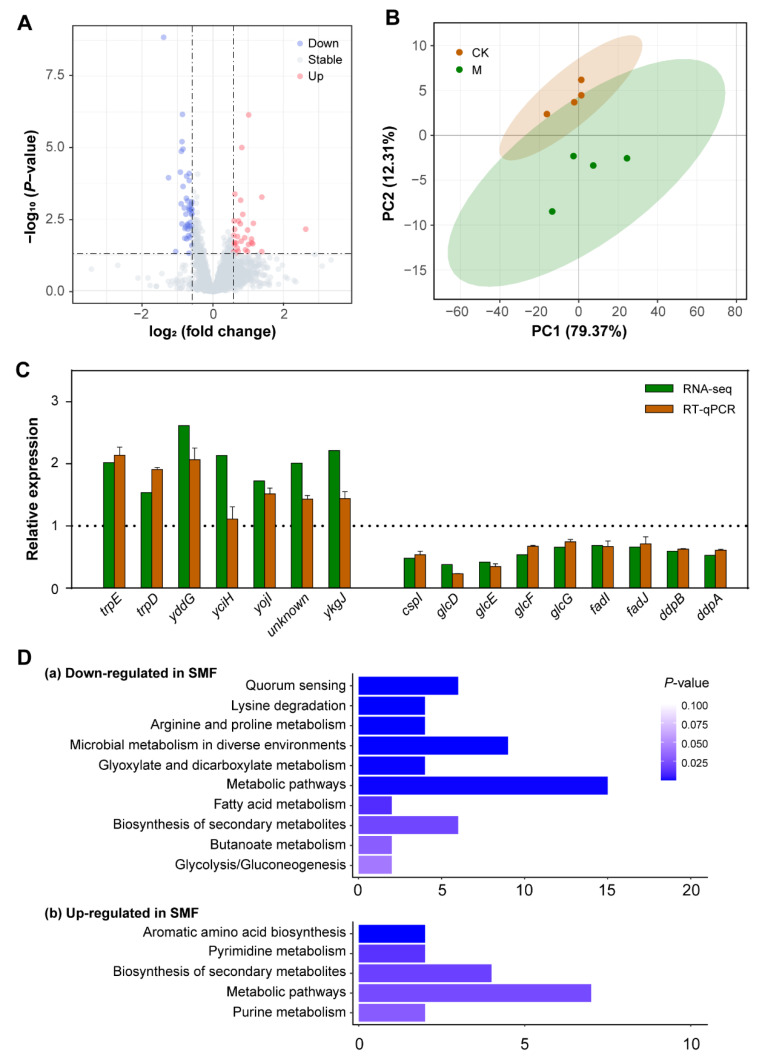
Transcriptomic analysis of 250 mT SMF effects on *E. coli* gene expression. (**A**) Volcano map of differentially expressed genes (DEGs) between SMF-treated and -untreated groups. Statistically upregulated and downregulated DEGs are shown as red or blue dots, respectively; the genes without significant difference in expression are shown as gray dots. (**B**) PCA analysis of the samples using transcriptomic data from four independent biological replicates. (**C**) Validation of RNA-Seq data by RT-qPCR. (**D**) KEGG analysis of SMF effects based on DEGs.

**Figure 3 cells-11-00827-f003:**
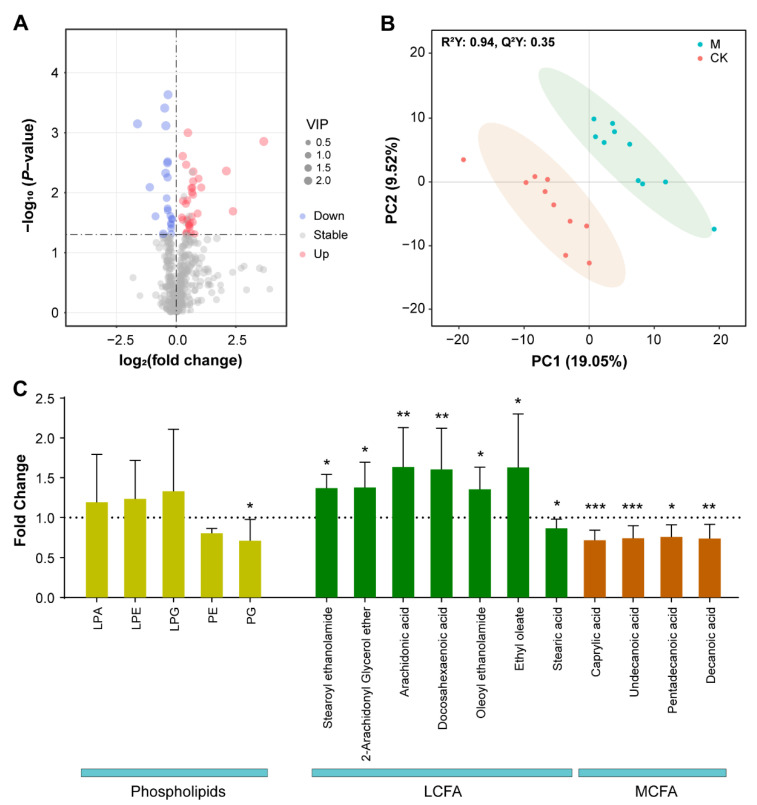
Metabolomic analysis of 250 mT SMF effects on *E. coli* metabolites. (**A**) Volcano map of the identified metabolites. Upregulated and downregulated metabolites with significant difference between SMF-treated and -untreated groups are shown as red and blue dots. Metabolites with no significant change are shown as gray dots. The size of dot represents the variable importance in projection (VIP). (**B**) PLS-DA score plots of the samples using metabolome data from ten independent biological replicates. (**C**) Effect of SMF on contents of phospholipids, LCFA and MCFA. Data shown as mean ± SD. Ten independent biological repeats were analyzed. Statistical analysis was performed by *t*-test. * *p* < 0.05, ** *p* < 0.01, *** *p* < 0.001.

**Figure 4 cells-11-00827-f004:**
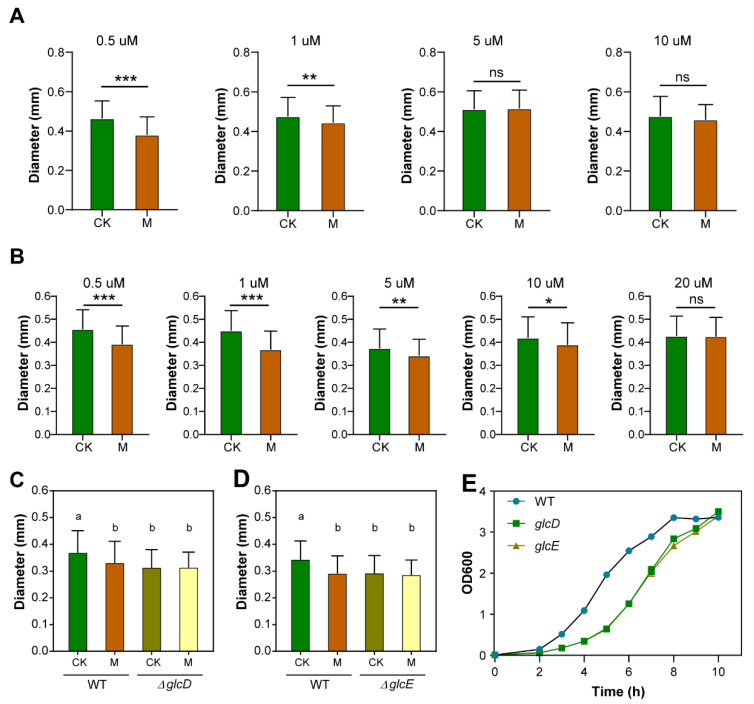
Effect of glycolate oxidase on SMF-inhibited *E. coli* growth. (**A**) Colony diameter of *E. coli* cultured on solid LB medium with various glyoxylate levels. (**B**) Colony diameter of *E. coli* cultured on solid LB medium with various glycolate levels. (**C**) Effect of *glcD* mutations on colony diameter of *E. coli*. (**D**) Effect of *glcE* mutations on colony diameter of *E. coli*. (**E**) Growth curves of *E. coli* TOP10 strain (WT) and *glcD*, *glcE* mutant strains under GMF condition. Data shown as mean ± SD. *n* > 50. Statistical analysis was performed by *t*-test. ^ns^
*p* > 0.05, * *p* < 0.05, ** *p* < 0.01, *** *p* < 0.001. Lowercase letters on the top of bar plots in (**C**,**D**) indicate significantly different groups as determined by one-way ANOVA with post-hoc Tukey HSD testing (*p* < 0.001).

**Figure 5 cells-11-00827-f005:**
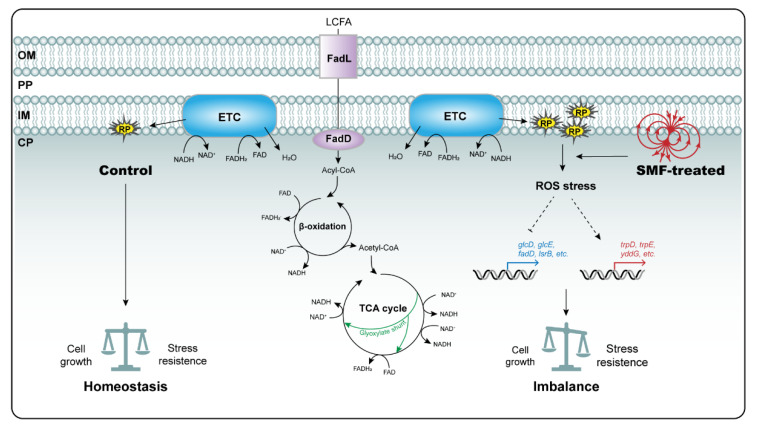
A model proposed to explain the biological SMF effects on *E. coli* growth. The detailed explanation is described in the text. Upregulated genes were shown with red line and downregulated genes were shown with blue line. LCFA: long-chain fatty acids; OM: outer membrane; PP: periplasm; IM: Inner membrane; CP: cytoplasm; ETC: electron transport chain; RP: radical pairs.

## Data Availability

The data used to support the findings of this study are included within the article.

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
