# Peer review of "Static Magnetic Field Inhibits Growth of Escherichia coli Colonies via Restriction of Carbon Source Utilization"

_cells, 2022, doi:10.3390/cells11050827_

Round 1

Reviewer 1 Report

This manuscript investigated the effects of static magnetic field (SMF) on E. coli cells. The authors found that 250 mT SMF inhibited the growth of E. coli colonies, and the inhibition of bacterial growth was attributed to the disturbance of carbon source utilization by transcriptomic and metabolomic analyses. The paper provides some interesting results. I have some comments or suggestions which need to be addressed by the authors.

1. The intensity and distribution of SMF is important to biological experiment. A distribution map or the uniformity of magnetic field was needed. In addition, the effects due to the different uniformity of the magnetic fields at 80 mT, 160 mT, 210 mT and 250 mT should be discussed.

2. Line 9, the magnetic field used by repetitive transcranial magnetic stimulation in ref [8] is not SMF.

3. Line 43, magnetosome should be “nano-scale crystal”.

4. Please show the detailed information of Crispr/Cas9 technology to make ΔglcD and ΔglcE in the TOP10.

5. Please add the detailed materials and methods to indicate how to get Figure 4E.

6. Spelling and format checks are strongly recommended.

(1) Please check if any typo in the text, such as “CUFs” (line14 and 221); “MSF” (line395); “TAC” (line 454).

(2) Please show the full name of KEGG when first appeared in the text. (line 182, not line259)

(3) Please show the full name of MSG. (line441)

(4) Please check Figure 4 caption. Is (A) reversed with (B)? The meaning of “a” and “b” in (C) and (D) should be introduced.

(5) Line 421, “260 mT SMF group”?

(6) Journal title in ref[10] was lost.

Author Response

Point 1: The intensity and distribution of SMF is important to biological experiment. A distribution map or the uniformity of magnetic field was needed. In addition, the effects due to the different uniformity of the magnetic fields at 80 mT, 160 mT, 210 mT and 250 mT should be discussed.

Response 1: Thank you for the comment. We added the data about the intensity and distribution of the used magnet, and a simple discussion about the magnetic efficiency under different intensities of SMF. With regard to different effects caused by different magnetic field intensities, since SMF intensity used in this study was much higher than that of the GMF, it is reasonable to assume that bacteria were grown under a stress condition. Under our condition, treatment with 160 mT or 80 mT SMF had no significant effect on bacterial growth, suggesting that bacteria can resist a certain level of SMF stress. Thus, we propose that there is a threshold of SMF intensity for biological effects. However, molecular mechanism underlying the SMF-inhibited bacterial growth needs to be investigated in the future.

Point 2: Line 9, the magnetic field used by repetitive transcranial magnetic stimulation in ref [8] is not SMF.

Response 2: Thanks for your careful reading. We deleted this inappropriate citation.

Point 3: Line 43, magnetosome should be “nano-scale crystal”.

Response 3: Thank you for your correction. We revised the word as you suggested.

Point 4: Please show the detailed information of Crispr/Cas9 technology to make ΔglcD and ΔglcE in the TOP10.

Response 4: Thanks for your comment. We added the required information in the part of materials and methods.

Point 5: Please add the detailed materials and methods to indicate how to get Figure 4E.

Response 5: Thanks for your comment. We added the method how to measure the growth rate of the bacteria.

Point 6: Spelling and format checks are strongly recommended.

(1) Please check if any typo in the text, such as “CUFs” (line14 and 221); “MSF” (line395); “TAC” (line 454).

(2) Please show the full name of KEGG when first appeared in the text. (line 182, not line259)

(3) Please show the full name of MSG. (line441)

(4) Please check Figure 4 caption. Is (A) reversed with (B)? The meaning of “a” and “b” in (C) and (D) should be introduced.

(5) Line 421, “260 mT SMF group”?

(6) Journal title in ref[10] was lost.

Response 6: Thank you very much for your corrections. We corrected the above mistakes, and carefully checked spelling and format.

Reviewer 2 Report

The manuscript presents the results on the application of 250 mT static magnetic field to E. coli colony-forming units. The results are interesting and the manuscript is suitable for publication in Cells.  There are few suggestions to improve the manuscript:

  1. Introduction is too long and lacks specifics. There are lengthy examples before mentioning the mechanism of magnetic field effect (ion-radical pair (IRP) in the singlet state with the antiparallel spin, which can be transformed into a triplet state with the parallel spin by external magnetic fields). Stating that “it remains unclear to what extent biological effects of SMF can be explained through the enzymatic mechanism”the authors continue with examples which does not appear to clarify the mechanisms. More over, the work output is aimed at “providing new insights into the mechanism for E. coli to adapt to the moderate intensity SMF”, but I cannot find the explanation of new mechanisms. In fact, only the radical pair mechanism was discussed in Section 4. Therefore, the corresponding parts of the manuscript should be modified.

  1. Figure 5. Only right-hand parts of the two sketches are different. The same parts are better to combine and show the deference related to RP.
  2. Lines 83-85- meaning is unclear
  3. Abbreviations

There are many abbreviations used. In this case,  a list of abbreviations could be helpful.

It seems some abbreviations were not explained (LB liquid, intracellular ATP).

  1. English should be polished. There are quite a few grammar issues.

Example:

Lines 85-86 “In general, it is presumed that the higher SMF  strength the more harmful to microbial growth and survival.”

Proposed:

In general, it is assumed that the higher the strength of the SMF, the worse the conditions for microbial growth and survival.

Lines 522-523

while AI-2 accumulates extracellularly and reaches maximally in the  exponential phase, and then decreases rapidly in the stationary phase.

Proposed:

“…while AI-2 accumulates extracellularly and peaks in the exponential phase and then declines rapidly in the stationary phase

Author Response

Point 1: Introduction is too long and lacks specifics. There are lengthy examples before mentioning the mechanism of magnetic field effect (ion-radical pair (IRP) in the singlet state with the antiparallel spin, which can be transformed into a triplet state with the parallel spin by external magnetic fields). Stating that “it remains unclear to what extent biological effects of SMF can be explained through the enzymatic mechanism”the authors continue with examples which does not appear to clarify the mechanisms. More over, the work output is aimed at “providing new insights into the mechanism for E. coli to adapt to the moderate intensity SMF”, but I cannot find the explanation of new mechanisms. In fact, only the radical pair mechanism was discussed in Section 4. Therefore, the corresponding parts of the manuscript should be modified.

Response 1: Thank you very much for the comment. We shortened the introduction part by deleting several examples. We also modefied our statement in a more appropriate manner “providing a metabolic basis for E. coli to adapt to the moderate intensity SMF”.

Point 2: Figure 5. Only right-hand parts of the two sketches are different. The same parts are better to combine and show the deference related to RP.

Response 2: Thanks for your suggestion. We revised the model as you proposed.

Point 3: Lines 83-85- meaning is unclear

Response 3: Thank you for your comment. We deleted this sentence.

Point 4: Abbreviations There are many abbreviations used. In this case,  a list of abbreviations could be helpful. It seems some abbreviations were not explained (LB liquid, intracellular ATP).

Response 4: Thanks for your comment. We added the whole names for abbreviations according to the journal guideline.

Point 5: English should be polished. There are quite a few grammar issues.

Example:

Lines 85-86 “In general, it is presumed that the higher SMF  strength the more harmful to microbial growth and survival.”

Proposed:

In general, it is assumed that the higher the strength of the SMF, the worse the conditions for microbial growth and survival.

Lines 522-523

while AI-2 accumulates extracellularly and reaches maximally in the  exponential phase, and then decreases rapidly in the stationary phase.

Proposed:

“…while AI-2 accumulates extracellularly and peaks in the exponential phase and then declines rapidly in the stationary phase.

Response 5: Thank you very much for your revision. We carefully revised the manuscript.